EMBO
Molecular Medicine

# Human ALPI deficiency causes inflammatory bowel disease and highlights a key mechanism of gut homeostasis

Marianna Parlato[1,2], Fabienne Charbit-Henrion[1,2,3,4], Jie Pan[5], Claudio Romano[2,6], Rémi Duclaux-Loras[1,2,3], Marie-Helene Le Du[7], Neil Warner[5], Paola Francalanci[8], Julie Bruneau[3,9], Marc Bras[10], Mohammed Zarhrate[11], Bernadette Bègue[1,2], Nicolas Guegan[1,3], Sabine Rakotobe[1,2], Nathalie Kapel[12], Paola De Angelis[8], Anne M Griffiths[5], Karoline Fiedler[5] [ID], Eileen Crowley[5], Frank Ruemmele[1,2,3,4], Aleixo M Muise[5,13,14,†,*] & Nadine Cerf-Bensussan[1,2,3,†,**] [ID]

## Abstract

Herein, we report the first identification of biallelic-inherited mutations in *ALPI* as a Mendelian cause of inflammatory bowel disease in two unrelated patients. *ALPI* encodes for intestinal phosphatase alkaline, a brush border metalloenzyme that hydrolyses phosphate from the lipid A moiety of lipopolysaccharides and thereby drastically reduces Toll-like receptor 4 agonist activity. Prediction tools and structural modelling indicate that all mutations affect critical residues or inter-subunit interactions, and heterologous expression in HEK293T cells demonstrated that all *ALPI* mutations were loss of function. *ALPI* mutations impaired either stability or catalytic activity of ALPI and rendered it unable to detoxify lipopolysaccharide-dependent signalling. Furthermore, ALPI expression was reduced in patients' biopsies, and ALPI activity was undetectable in ALPI-deficient patient's stool. Our findings highlight the crucial role of ALPI in regulating host–microbiota interactions and restraining host inflammatory responses. These results indicate that *ALPI* mutations should be included in screening for monogenic causes of inflammatory bowel diseases and lay the groundwork for ALPI-based treatments in intestinal inflammatory disorders.

**Keywords** inflammatory bowel diseases; intestinal phosphatase alkaline; monogenic disease
**Subject Categories** Digestive System; Genetics, Gene Therapy & Genetic Disease; Metabolism

## Introduction

Inflammatory bowel diseases (IBDs) are complex, and severe disorders ascribed to alterations in the dialogue between the microbiota and the host immune system (Bouma & Strober, 2003; Maloy & Powrie, 2011). Insights into pathogenesis, first derived from animal studies, have highlighted how multiple mechanisms cooperate to build an efficacious and tightly regulated intestinal barrier that enables hosts to cope with the complex and diverse community of microbes inhabiting the bowel (Bouma & Strober, 2003; Maloy & Powrie, 2011). In humans, the common polygenic forms of IBDs have been thoroughly investigated using genomewide association studies (GWAS). These studies demonstrated that variants affecting

---

1   INSERM, UMR1163, Laboratory of Intestinal Immunity and Institut Imagine, Paris, France
2   GENIUS group from ESPGHAN
3   Université Paris Descartes-Sorbonne Paris Cité, Paris, France
4   Department of Pediatric Gastroenterology, Assistance Publique-Hôpitaux de Paris, Hôpital Necker-Enfants Malades, Paris, France
5   SickKids Inflammatory Bowel Disease Center and Cell Biology Program, Research Institute, Hospital for Sick Children, Toronto, ON, Canada
6   Unit of Pediatrics, Department of Human Pathology in Adulthood and Childhood "G. Barresi", University of Messina, Messina, Italy
7   Department of Biochemistry, Biophysics and Structural Biology, Institute for Integrative Biology of the Cell (I2BC), CEA, UMR 9198 CNRS, Université Paris-Sud, Gif-sur-Yvette, France
8   Digestive Endoscopy and Surgery Unit and Pathology Unit Bambino Gesù Children Hospital, IRCCS, Rome, Italy
9   Department of Pathology, Necker-Enfants Malades Hospital, Assistance Publique-Hôpitaux de Paris, Paris, France
10  Bioinformatics Platform, Université Paris-Descartes-Paris Sorbonne Centre and Institut Imagine, Paris, France
11  Genomic Platform, INSERM, UMR1163, Imagine Institute, Paris Descartes-Sorbonne Paris Cite University, Paris, France
12  Department of Functional Coprology, Pitié Salpêtrière Hospital, Assistance publique-Hôpitaux de Paris (AP-HP), Paris, France
13  Division of Gastroenterology, Hepatology, and Nutrition, Department of Pediatrics, Hospital for Sick Children, University of Toronto, Toronto, ON, Canada
14  Department of Biochemistry, Institute of Medical Science, University of Toronto, Toronto, ON, Canada
    *Corresponding author. Tel: +1 416 813 7735; E-mail: aleixo.muise@sickkids.ca
    **Corresponding author. Tel: +33 1 4275 4288; E-mail: nadine.cerf-bensussan@inserm.fr
    †These authors contributed equally to this work

---

expression or function of genes involved in innate defence or cell homeostasis may predispose to or protect against intestinal inflammation (Jostins *et al*, 2012). Recent advances in next-generation sequencing methods have enabled the identification of an expanding number of monogenic forms of IBDs generally characterized by very early onset and extreme severity (Uhlig, 2013; Uhlig & Muise, 2017). Genetic dissection of these rare disorders is indispensable to identify as early as possible the best therapeutic option and to improve long-term outcome and life quality. It also provides novel insight into the non-redundant mechanisms that are necessary to maintain mutualistic host–microbiota interactions in the gut.

Sensing of commensal bacteria by Toll-like receptors (TLRs) is critical to maintain intestinal epithelial homeostasis and to protect against direct injury (Rakoff-Nahoum *et al*, 2004), while hyperactive TLR signalling can foster intestinal inflammation and autoimmunity in many experimental settings (Asquith *et al*, 2010). In the gut, TLR4 is expressed by both epithelial and immune cells and can be activated by lipopolysaccharides (LPS), a family of complex glycolipids that are major constituents of the outer membrane of Gram-negative bacteria. In order to cope with the huge load of LPS released in the intestinal lumen by the microbiota and to restrain TLR4-dependent inflammatory signals, hosts have evolved several regulatory mechanisms. Animal studies have shown that one crucial mechanism is intestinal alkaline phosphatase (ALPI), a brush border metalloenzyme that catalyses phosphate hydrolysis of the lipid moiety of LPS and thereby drastically reduces LPS pro-inflammatory activity (Schromm *et al*, 1998; Goldberg *et al*, 2008).

Herein, using whole-exome sequencing (WES), we report the identification of *ALPI* mutations in two unrelated patients displaying severe intestinal inflammation and autoimmunity. Our data highlight how LPS detoxification by ALPI plays a crucial role in humans to restrain inflammatory responses to the microbiota. Moreover, our findings provide a genetic-based rationale for ALPI oral therapy in IBD.

## Results

### Clinical description of the patients

Two patients, P1 and P2, from two unrelated kindreds were investigated. P1 was born at term to non-consanguineous parents living in Italy. He presented at 2 years of age with severe diarrhoea and weight loss. Histology showed increased duodenal intraepithelial lymphocytes with mild villous atrophy (Fig 1A). Since serum IgA and IgG anti-tissue transglutaminase 2 (anti-TG2) titres were over 100 U/ml and HLA typing revealed DR3-DQ2/DR7-DQ8 haplotypes, he was diagnosed with coeliac disease (Fig EV1). Gluten-free diet for 1 year resolved his clinical symptoms and lowered TG2 titres. He was again referred to the clinic at 3 years of age because of recurrent abdominal pain, rectal bleeding and severe diarrhoea. Repeated endoscopic evaluations showed pancolitis with continuous inflammation and ulcerations of severe intensity from transverse colon to rectum. Mucosa was normal in the terminal ileum. Histological lesions combined dense lymphoplasmacytic infiltrate, mucosal ulcerations and erosions, mucin depletion, Paneth cell metaplasia and diffuse thickening of the muscularis mucosae in all colonic biopsies. Of note, no epithelioid granulomas were identified. Immunological workup showed normal serum concentrations of IgG, IgA, IgE and IgM and negative Epstein–Barr virus and cytomegalovirus serology, but the presence of cytoplasmic anti-neutrophil cytoplasmic (cANCA) antibodies. Biological evaluation showed elevated faecal calprotectin (200 μg/g). He was initially treated with steroids and azathioprine. In view of continued disease activity and steroid dependence, infliximab was initiated with no significant improvement. Due to lack of response, he underwent colectomy with ileorectal anastomosis. No extra-gastrointestinal manifestation was reported. A first screen using a targeted sequencing of 66 genes known to be associated with intestinal disorders did not detect any possibly damaging rare (frequency < 1%) variants in public or in-house databases (Uhlig *et al*, 2014).

P2, a boy born to non-consanguineous parents of Jewish Ashkenazi origin, initially presented at the age of 15.4 years with non-painful swelling of an ankle and hip. Family history revealed psoriatic arthritis in a maternal uncle and ulcerative colitis in the maternal grandmother. P2 was HLA-B27 negative but had anti-nuclear antibodies (titre 1:320). He was treated with non-steroidal anti-inflammatory drugs without improvement. Six months later, he became acutely sick with vomiting and abdominal pain, biological evidence of severe intestinal inflammation [white cell counts 16 (4–10 × $10^9$/l), neutrophils 12 (2–7.5 × $10^9$/l), platelets 891 (150–400 × $10^9$/l), CRP 127.5 (0–8 mg/l), ESR 93 (1–10 mm/h)], thickening of small and large bowel wall at ultrasound examination,

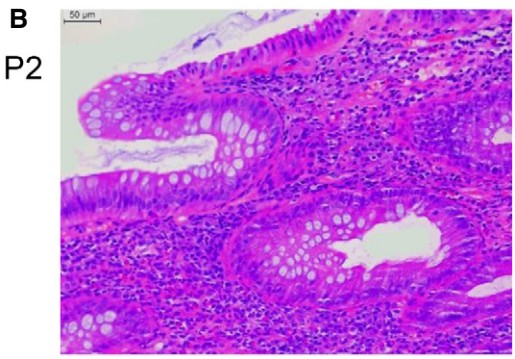

**Figure 1. Colonic inflammation in ALPI-deficient patients.**

A, B    Haematoxylin–eosin staining of colonic biopsies from P1 (A) and P2 (B). Scale bars: 100 μm (A) and 50 μm (B).

abnormal distal ileum at barium contrast imaging, endoscopic and histological evidence of mild inflammation in stomach and duodenum and of more extensive inflammation in the distal intestine with ulcerations of the ileocaecal valve and of the colon hepatic flexure, inflamed granulation tissue with a single granuloma in the ascending colon and one ulcer in sigmoid (Fig 1B). Serum IgM was normal, serum IgG and IgA increased (21.2 and 4.9 g/l, respectively), and anti-TG antibodies were negative. P2 was diagnosed with ileocolonic Crohn's disease and was first treated by exclusive enteral nutrition. Due to iterative clinical symptoms of partial small bowel obstructions, he received corticosteroids and anti-TNF therapy, and finally underwent ileocaecal resection, which showed moderate-to-severe chronic active ileitis with transmural chronic inflammation, fissure-type ulcerations, serosal fibrosis but no evidence of granulomas. After a few months on metronidazole, treatment could be discontinued without digestive relapse. His joint symptoms have resolved post-ileocaecal resection, and he is no longer on any rheumatological medications.

### Identification of heterozygous compound mutations in *ALPI* by WES

Whole-exome sequencing was performed on trios including affected individuals and their unaffected parents. Annotated data were analysed with in-house software. WES of patient 1 (and parents) resulted in > 144× the mean depth of coverage of the exonic targeted bases covered by a minimum of 30 independent reads and identified 101,319 variants. For P2, exome sequencing coverage was 20× or greater for > 85% of the bases targeted and identified 106,869 variants. Polymorphisms reported in public databases with minor allele frequency (MAF) > 1%, and synonymous variants were filtered out. Potential pathogenicity of each protein-coding variant was evaluated using OMIM (Online Mendelian Inheritance in Man), evolutionary conservation and prediction tools (SIFT, PolyPhen-2.2.2, Mutation Taster). Mutations were next ranked by combined annotation-dependent depletion (CADD) and compared with the mutation significance cut-off (MSC), a gene-level-specific cut-off for CADD scores (Itan *et al*, 2016). This pipeline did not identify any causal variants in known disease-relevant genes (Appendix Tables S1 and S2). Instead, it led to the identification of compound heterozygous mutations in the *ALPI* gene, encoding intestinal alkaline phosphatase, in both patients (Fig 2A and B, Table 1). Each variant was confirmed by Sanger sequencing (Figs 2C and EV2) with the segregation within each family being consistent with autosomal recessive inheritance. *ALPI* has eleven exons spanning 4.6 kb (UCSC Genome Browser hg37, chr2:233, 320, 822–233, 325, 455) and encodes for a 528 amino acid (aa) precursor protein featuring a signal peptide at its N-terminus (NT; 1–19 aa) and a C-terminal (CT) phosphatidyl-inositol glycan (GPI) anchor signal (504–528 aa). Active ALPI consists of two identically processed subunits (each lacking the first 19 and last 24 aa) bound to the cell surface via a post-translationally added GPI anchor (Fig 2D). All mutations were designated as disease causing by the majority of *in silico* predictions tools, with three variants resulting in the substitution of residues highly conserved across species (A97T, A350V and A360; Fig EV2) and one variant (Q439X) introducing a premature stop codon. Moreover, the combined annotation-dependent depletion (CADD) scores of each variant were well above the mutation significance cut-off

(MSC) of 3.13 defined for *ALPI* (Table 1). All mutations were either absent or present in only a heterozygous state at a frequency of less than 0.001% in the Exome Aggregation Consortium (ExAC) database (Table 1; Lek *et al*, 2016). Overall, *in silico* analyses suggested that the four variants identified in the two patients were not irrelevant polymorphisms but rare pathogenic mutations.

### Impaired expression of ALPI A97T and Q439X mutants

To assess the impact of the mutations on protein stability of ALPI, HEK293T cells were stably transduced with lentiviral particles encoding WT and mutant proteins. In parallel, and in order to define structure–function correlation, we generated a homology model of human ALPI based on the crystal structure of human placental alkaline phosphatase (ALPP), which shares 87% identity with ALPI (Le Du *et al*, 2001; Appendix Fig S1). We then analysed how mutations might affect the 3D structure of the ALPI homodimer using ConSurf server (Ashkenazy *et al*, 2016).

Immunoblotting (Fig 3A) and cell surface flow cytometry (Fig 3B) revealed significantly less ALPI protein in HEK293T cells stably transduced with A97T mutant than with the WT construct despite comparable mRNA expression (Fig EV3). Since the highly conserved alanine 97 is located at the β-strand in the dimer interface with each alanine facing the other (Fig 3C), we posited that substitution by threonine with its significantly larger, polar projecting side chain might impair protein dimerization and therefore stability. As expected, given the lack of a GPI anchoring site at the CT domain, the Q439X truncated protein was undetectable at the cell surface by flow cytometry (Fig 3B). However, a weak band with the predicted size of the truncated Q439X mutant was detected by immunoblot (Fig 3A). Reduced abundance of the corresponding transcript (Fig EV3) suggested nonsense-mediated decay. Since the 3D model predicts that each CT region is deeply buried at the interface between the two monomers and engaged in non-covalent interactions, the lack of CT might also impair proper folding and dimerization of the truncated Q439X mutant and affect protein stability (Fig 3D, centre). In contrast, A350V and A360V mutants yielded comparable amount of protein as the WT allele, as assessed by immunoblotting and flow cytometry (Fig 3A and B).

### Impaired function of ALPI A350V and A360V mutants

We next sought to determine the impact of mutations on protein function by performing *in vitro* phosphatase activity assays using para-nitrophenyl phosphate (pNPP) as a chromogenic substrate. In keeping with their reduced protein expression, A97T and Q439X mutants showed low or no residual phosphatase activity (Fig 3D). As indicated above, A350V and A360V mutations did not affect ALPI expression. Modelling of the ALPP structure further suggested that neither mutation directly altered the predicted catalytic site (Ser111; Fig 3E, bottom). However, both modified residues are located on the same α-helix in the folding core surrounding the catalytic site, suggesting that the amino acid substitutions may destabilize the overall shape of the phosphatase domain, and thus indirectly affect metal chelation and/or enzyme activity. In support of this hypothesis, the enzymatic activity of A350V and A360V mutants was significantly reduced to 10 and 30% of WT ALPI activity, respectively (Fig 3F).

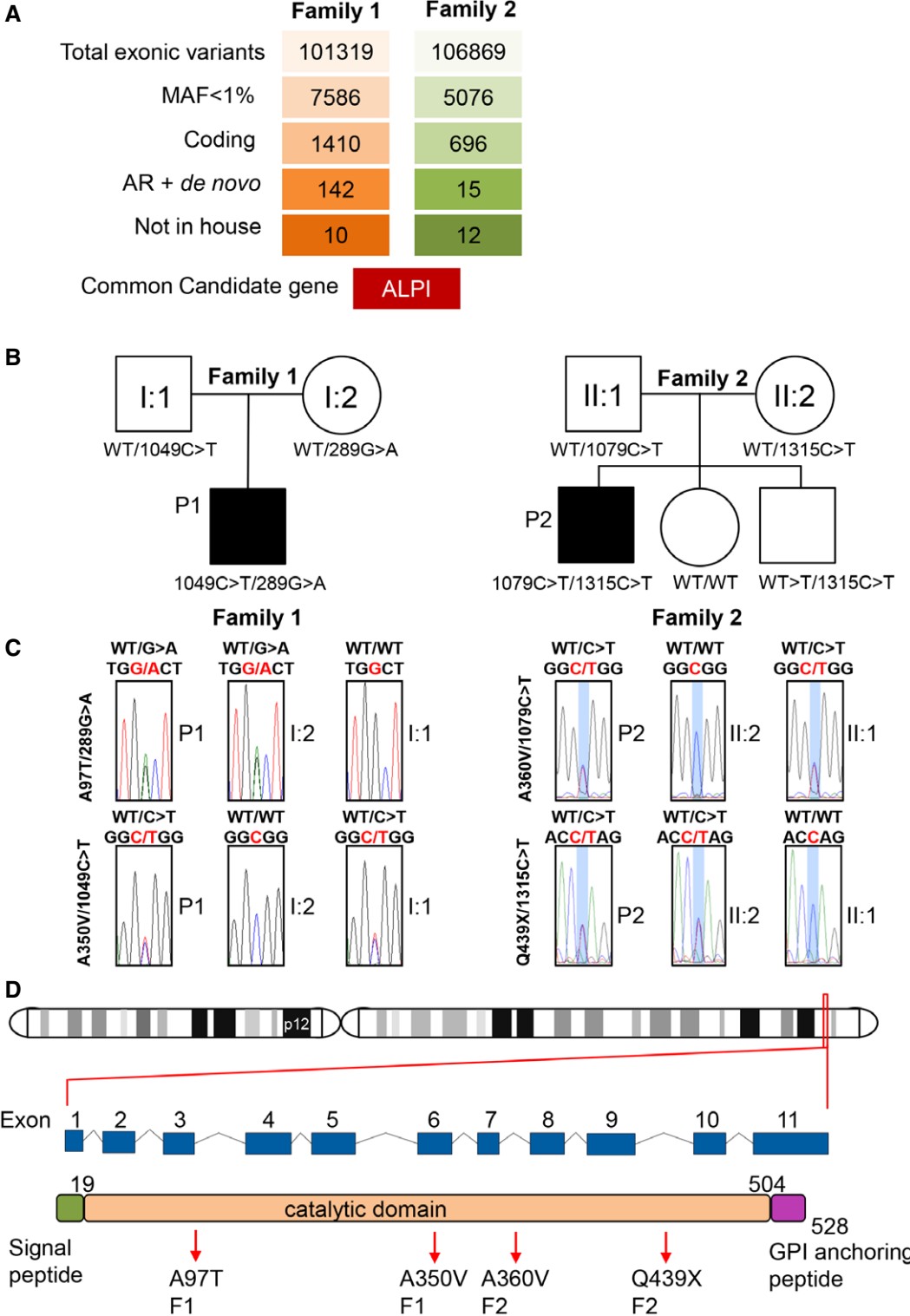

**Figure 2. WES identification of heterozygous compound mutations in *ALPI*.**

A   Variant identification pipeline for WES in P1 and P2 (see also Appendix Tables S1 and S2).

B   Unrelated familial trees showing affected children in black and healthy individuals in white.

C   Sanger sequencing of the region corresponding to mutations in ALPI in both families.

D   Location of *ALPI* on chromosome 2 and diagrams featuring *ALPI* gene with its nine exons and ALPI protein with its N-terminal signal peptide, phosphatase domain and C-terminal recognition signal for the transamidase complex (GPI-anchor attachment site), which removes the GPI signal sequence and replaces it by a preformed GPI precursor glycolipid. Arrows point to mutations identified in P1 and P2. Numbers indicate amino acid position.

**Table 1. Features of variants identified in P1 and P2.**

| Patient | Exon | Position | | | MAF | Prediction | | | |
| | | Genomic Chr2 (GRCh37) | cDNA NM_001631.4 | Proteins | | CADD MSC:3.13 | PoliPhen-2 | SIFT | Mutation Taster |
|---|---|---|---|---|---|---|---|---|---|
| 1 | 3 | g.233321394G>A | c.289G>A | p.Ala97Thr | 0.0004362 | 27.1 | Probably damaging score: 1.0 | Deleterious score: 0.03 | Disease causing P-value: 0.997 |
| | 9 | g.233322984C>T | c.1049C>T | p.Ala350Val | No Freq | 19.95 | Possibly damaging score: 0.532 | Tolerated score: 0.35 | Disease causing P-value: 0.644 |
| 2 | 9 | g.233323014C>T | c.1079C>T | p.Ala360Val | 0.0006363 | 27.8 | Probably damaging score: 0.990 | Tolerated score: 0.36 | Disease causing P-value: 1 |
| | 11 | g.233323584C>T | c.1315C>T | p.Gln439X | 0.0001295 | 19.3 | – | – | – |

MAF, minor allelic frequency based on 60,706 individuals genotyped as part of the Exome Aggregation Consortium (ExAC: http://exac.broadinstitute.org). Variant predictions are based on PolyPhen-2 (http://genetics.bwh.harvard.edu/pph2), SIFT (http://sift.jcvi.org), Mutation Taster (http://www.mutationtaster.org). Mutation significance cut-off (MSC), a gene-level-specific cut-off for CADD scores, was generated for *ALPI* (http://pec630.rockefeller.edu:8080/MSC/).

ALPI has been shown to detoxify LPS molecules by removing a phosphate group from its lipid A moiety. The resulting monophosphorylated LPS retains only weak TLR4 agonist activity (Schromm *et al*, 1998; Bates *et al*, 2007; Goldberg *et al*, 2008). As the mutants identified here were all loss-of-function alleles, we hypothesized that treatment of LPS with ALPI mutants would not inhibit LPS immunostimulatory activity. Accordingly, and in contrast with the WT allele, none of the ALPI mutants could significantly inhibit LPS-dependent IL-8 transcription in THP1 monocytic cells (Fig 3G).

### *In vivo* assessment of ALPI mutations

ALPI is mainly expressed in the small intestinal brush border, from which it is released into the lumen within vesicles that transport active ALPI towards distal intestinal sites (Eliakim *et al*, 1991; Bates *et al*, 2007; Tuin *et al*, 2009; Shifrin *et al*, 2012; Malo, 2015). In keeping with *in vitro* evidence that ALPI mutations affect protein expression, immunofluorescence analysis of small intestinal biopsies showed markedly reduced ALPI staining in P1 and P2 compared with histologically normal or IBD controls (Fig 4A and B). In ALPI-deficient mice, alternative isoforms of alkaline phosphatase (AP) are upregulated (Narisawa *et al*, 2007). Similarly, ALPI-deficient patients displayed enhanced expression of tissue-non-specific alkaline phosphatase (TNAP), the other AP isoenzyme present in human gut mucosa (López-Posadas *et al*, 2011). In contrast with ALPI, TNAP staining largely predominated in *lamina propria* in controls and in ALPI-deficient patients. Yet, P1 and P2 also displayed some epithelial expression, suggesting that TNAP might compensate for ALPI deficiency in the intestinal lumen (Fig EV4). To address this question, AP activity was assessed in stool in the presence of L-phenylalanine (L-Phe), a specific inhibitor of ALPI, or L-homoarginine (L-Arg), a specific inhibitor of TNAP. As shown in Fig 4C, stools of non-inflamed controls displayed substantial AP activity, which was largely inhibited by L-Phe but not by L-Arg. This result was consistent with previous reports indicating that most AP activity in stools is due to ALPI with some residual AP activity of microbial origin (Malo, 2015; Fig 4C). ALPI activity was significantly reduced in stools of

patients with intestinal inflammation (faecal calprotectin > 250 μg/g; Fig 4D), a result consistent with previous studies showing decreased ALPI expression in the inflamed intestine (Molnár *et al*, 2012a,b). However, TNAP remained, as in controls, undetectable. ALPI analysis could not be performed in P2, who refused stool sampling, but both ALPI and TNAP were undetectable in P1 stools sampled at time of remission 2 years after colectomy, confirming the loss-of-function mutations in ALPI and the lack of intraluminal substitution by TNAP in this patient.

## Discussion

Herein, we report for the first time an inherited deficiency in ALPI in two unrelated patients with severe intestinal inflammation refractory to medical treatment. Both patients carry compound heterozygous loss-of-function mutations that impair either stability or catalytic activity of ALPI. Accordingly, ALPI expression is markedly reduced in small intestinal biopsies, and its activity was undetectable in stools. ALPI is produced in the intestinal brush border and released in an active form into the intestinal lumen where it can dephosphorylate microbiota-derived LPS and thereby considerably reduces its TLR4 agonist activity. In keeping with loss of function, the mutated ALPI alleles could not inhibit LPS activity when measured *in vitro* by the induction of IL-8 (Fig 3G). Although TNAP, another AP expressed in the human gut mucosa, was upregulated in the intestinal mucosa of both ALPI-deficient patients, we found no evidence of its secretion in the intestinal lumen, a result consistent with previous results (Malo, 2015). We therefore conclude that LPS detoxification by ALPI in the intestinal lumen is severely impaired in P1 and P2. This conclusion is in keeping with recent work showing how reduced expression of ALPI induced by recurrent non-lethal *Salmonella* infections impairs intraluminal dephosphorylation/detoxification of LPS (Yang *et al*, 2017). Since only two cases of ALPI deficiency were identified, we suggest that this defect is instrumental in predisposing P1 and P2 to severe intestinal inflammation, and our conclusions are strongly supported by data derived from animal models (Malo *et al*, 2010; Ramasamy *et al*, 2011; Rentea

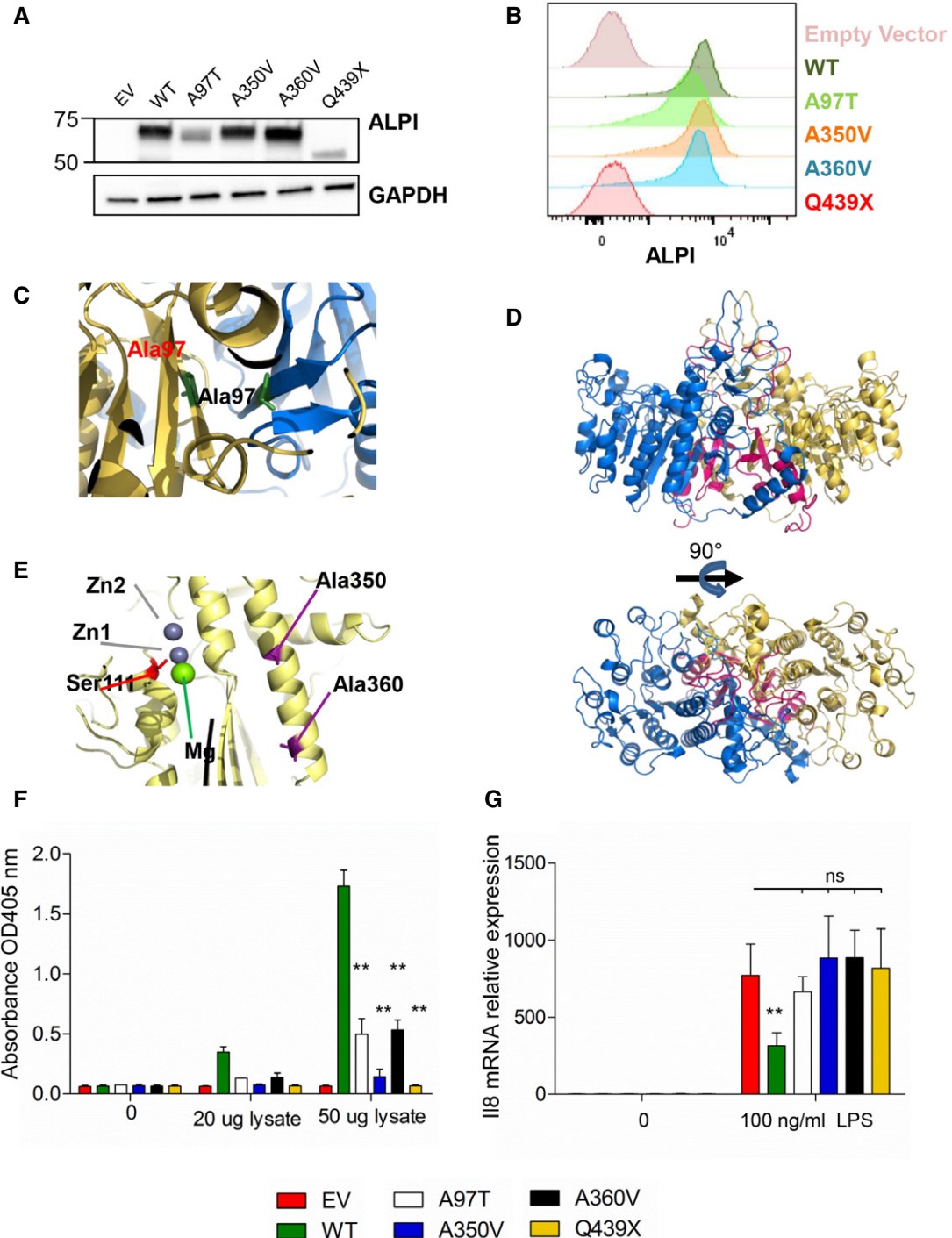

**Figure 3.    Loss of function of ALPI mutants.**

A, B    Analysis with ALPI antibody of HEK293T cells (lacking endogenous ALPI expression) after mock transduction or transduction with lentiviral particles encoding indicated mutants. (A) Western blot of whole-cell lysates. (B) Flow cytometry analysis of surface-labelled cells.

C–E    3D modelling of ALPI calculated on the basis of the crystal structure of ALPP. ALPI is represented with one monomer in yellow, and the second in blue. Alanine 350, alanine 360 and catalytic Ser111 are represented in stick in violet, cyan and red, respectively. Deletion of the C-terminal from Gln439 is shown in pink. Top and bottom views shown in (D) correspond to a 90° rotation.

F    pNPP phosphatase activity of WT and ALPI mutants measured by the OD405 nm of the reaction supernatants. $n = 5$, error bars indicate SD. **$P = 0.0079$, nonparametric, unpaired two-tailed Mann–Whitney test.

G    IL-8 inhibition of LPS-induced transcription in THP1 cells by WT and ALPI mutants. GAPDH, glyceraldehyde-3-phosphate dehydrogenase; EV, empty vector. $n = 3$, error bars indicate SD. **$P = 0.022$, nonparametric, unpaired two-tailed Mann–Whitney test.

Source data are available online for this figure.

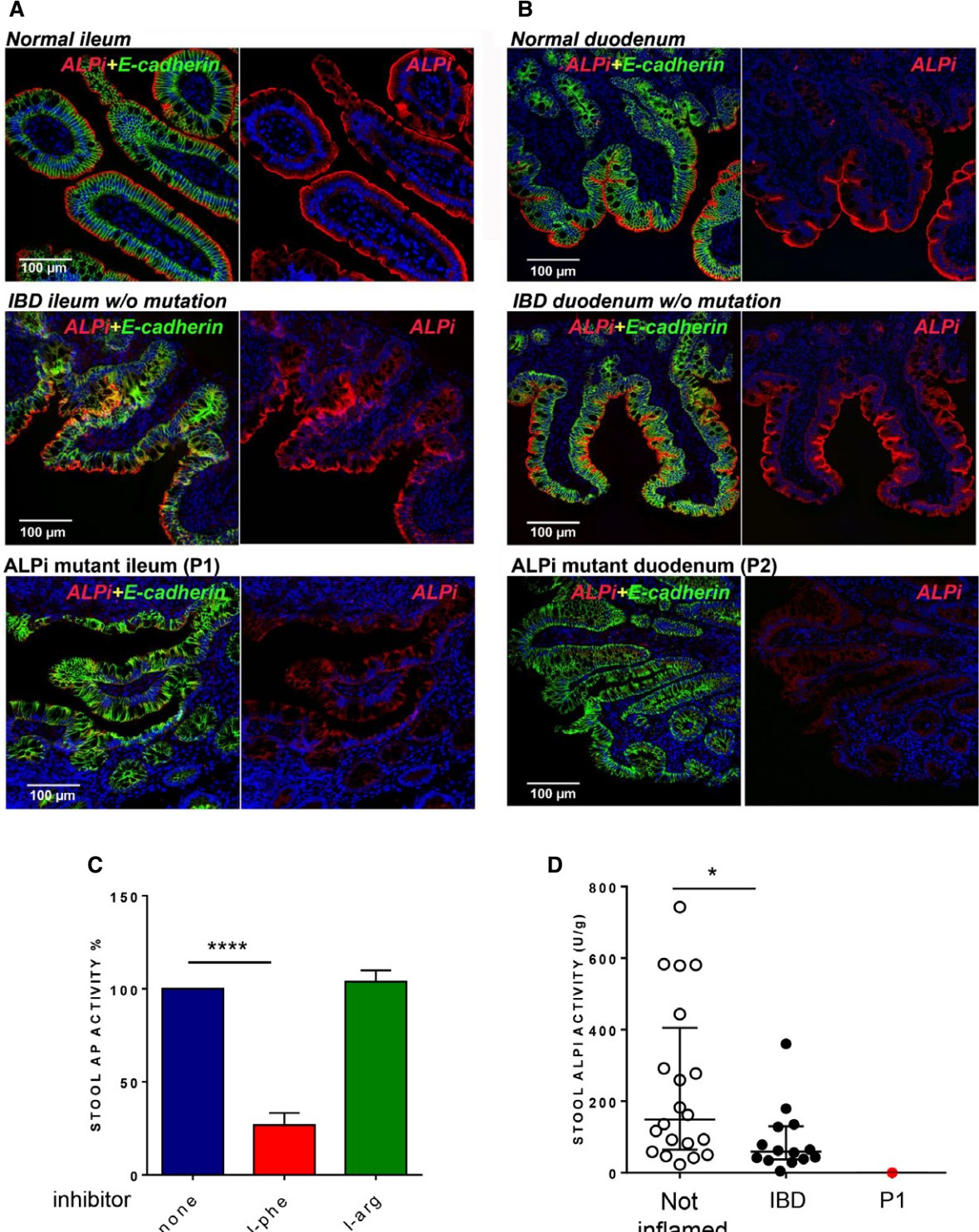

**Figure 4.   Reduced ALPI expression in patients' small intestinal biopsies.**

A, B    Immunofluorescence microscopy of duodenum (A) and ileum (B) sections from Patient 1 (A) and Patient 2 (B) compared to normal or IBD controls. Sections were stained with DAPI (4′,6-diamidino-2-phenylindole) for DNA (blue) and antibodies against ALPI alone (second column, red) or ALPI and E-cadherin (green) (first column, merge staining). Magnification 20× (scale bar = 100 μm).

C    Effect of AP inhibitors (L-phe, L-arg) on AP activity in stools of non-inflamed controls ($n$ = 20). Data are expressed as mean ± SD. ****$P$ < 0.0001, nonparametric, unpaired two-tailed Mann–Whitney test.

D    ALPI activity in stools of non-inflamed ($n$ = 20) or IBD controls ($n$ = 14) and of P1. AP values are expressed as units of ALPI/g stool and as medians with interquartiles; *$P$ = 0.0122, nonparametric, unpaired two-tailed Mann–Whitney test.

et al, 2012; Kaliannan et al, 2013; Fawley et al, 2017; Yang et al, 2017). Zebrafish knocked down for ALPI displayed markedly increased LPS-induced mortality upon oral challenge (Bates et al, 2007). Moreover, in the absence of ALPI, excessive numbers of neutrophils were recruited into the intestine of conventionally raised but not that of germ-free zebrafish, pointing to a key role for ALPI in suppressing pro-inflammatory responses to the microbiota. In mammals, there is ample evidence that continuous intestinal exposure to LPS induces the degradation of signalling molecules downstream of TLR4, so that, at steady state, epithelial cells and lamina propria mononuclear phagocytes are largely unresponsive to LPS (Smith et al, 2001; Lotz et al, 2006). Endogenous tolerance to LPS is broken upon induction of inflammation induced either by gut injury or infection, notably due to the recruitment of inflammatory monocytes into the intestinal mucosa (Kamada et al, 2008). ALPI-mediated detoxification may then become indispensable to curb inflammation. This scenario is coherent with data in ALPI-deficient mice, which did not spontaneously develop intestinal inflammation, but displayed markedly increased susceptibility to dextran sodium sulphate-induced colitis (Ramasamy et al, 2011) as well as increased gut permeability (Rentea et al, 2012) and enhanced LPS translocation when subjected to direct or indirect gut injury (Kaliannan et al, 2013). This scenario is also supported by data in mice exposed to recurrent non-lethal infection by Salmonella typhimurium. The mice are able to eradicate the bacteria but ultimately develop chronic colitis due to increased degradation of ALPI and lesser availability of active ALPI in the intestinal lumen (Yang et al, 2017). The necessity of an environmental trigger to uncover genetic susceptibility to intestinal inflammation in ALPI-deficient individuals might account for the distinct ages of P1 and P2 at disease onset, leading to P1 developing very early onset IBDs while P2 presented as a teenager. Of note, both patients developed autoantibodies and diseases often linked to autoimmunity just before or concurrent to distal intestinal inflammation. Thus, P1 developed coeliac disease, to which he was genetically predisposed by the HLA-DQ2 haplotype, while P2, who had an uncle with psoriatic arthritis, developed arthritis. Whether ALPI deficiency favoured autoimmunity-related manifestations in P1 and P2 remains uncertain. Indeed, the recent suggestion that microbiota enriched in Bacteroidetes species producing hexa-acylated LPS with strong TLR4 agonist activity promotes endotoxin tolerance and thereby protects against type I diabetes (Vatanen et al, 2016) seems contradictory. Removal of one of the two phosphorylated residues of LPS by ALPI may, however, be necessary to fine-tune TLR4 activation and to foster endotoxin tolerance. Accordingly, injection of monophosphorylated lipid A could attenuate LPS-induced inflammatory responses in healthy volunteers (Astiz et al, 1995).

Importantly, oral ALPI supplementation protected ALPI-deficient mice from increased susceptibility to DSS-induced colitis (Ramasamy et al, 2011) and from chronic colitis induced by recurrent non-lethal infections by S. typhimurium (Yang et al, 2017). This treatment also significantly reduced intestinal inflammation in several mouse models of colitis and prevented LPS translocation (Ramasamy et al, 2011; Martínez-Moya et al, 2012; Rentea et al, 2012; Heinzerling et al, 2014; Biesterveld et al, 2015). Moreover, a small trial in patients with ulcerative colitis showed that administration of exogenous ALPI was not only well tolerated but also associated with short-term improvement in clinical scores (Lukas et al, 2010). Altogether these data indicate that oral administration of recombinant ALPI is a very attractive therapeutic strategy in order to alleviate intestinal inflammation in ALPI-deficient patients. At the time of the present study, both P1 and P2 had undergone intestinal resection and were in long-term remission, circumventing the need for any further treatment. It is, however, tempting to speculate that if oral ALPI supplementation had been initiated early after disease onset, it might have prevented the development of refractory disease and the need for surgery. These observations stress the importance of early molecular diagnosis in patients with monogenic intestinal disorders in order to optimize therapy and improve long-term outcome and life quality.

In conclusion, our results support the view that ALPI deficiency is a novel inherited cause of intestinal inflammation and underscore the central and conserved role of ALPI in maintaining gut homeostasis. Moreover, observations in patients with monogenic ALPI deficiency provide strong rationale for oral ALPI treatment in common multi-factorial cases of IBDs in order to alleviate ileocolonic inflammation.

# Materials and Methods

### Patients

For genetic studies, patients were enrolled in two independent IBD paediatric cohorts in Hôpital Necker-Enfants Malades in Paris (P1) and in Hospital for Sick Children in Toronto (P2). Genetic studies were carried out in accordance with approved institutional protocols (CPP Ile-de-France II and the Research Ethics Board at the Hospital for Sick Children) and conform to the 1975 Declaration of Helsinki and the Department of Health and Human Services Belmont Report. Informed consent for genetic testing was obtained from all participants.

Dosage of ALPI activity was performed in the stools of 34 informed patients in parallel with calprotectin dosage performed for diagnosis purposes in Pitié-Salpêtrière Hospital. They were divided into two groups depending on calprotectin concentration: inflamed (calprotectin > 250 μg/g; n = 14; aged 13–60 years) and non-inflamed (calprotectin < 250 μg/g; n = 20; aged 2–66 years).

### Whole-exome sequencing and analysis

Genomic DNA was isolated from peripheral blood cells with QIAamp® DNA Blood Mini Kit (Qiagen, Courtaboeuf, France; P1 family) and from whole blood with Puregene Blood Core Kit (Qiagen, Germantown, MD; P2 family). DNA was processed for exome capture using the 58 Mb V6 version of the SureSelect Exome kit (Agilent Technologies, Les Ulis, France; P1 and parents) or the NimbleGen VCRome 2.1 design (Roche sequencing; P2 and parents). Paired-end sequencing generating, respectively, 100- and 75-base reads was performed on Illumina HiSeq 2500 (Illumina, San Diego, CA) at institute Imagine (P1) or at the Regeneron Genetics Center (RGC; P2). Sequences were aligned with the GRCh37 reference human genome using the Burrows-Wheeler Aligner, and downstream processing was carried out with the Genome Analysis Toolkit (GATK), SAMtools

and Picard, according to documented best practices (http://www.broadinstitute.org/gatk/guide/topic?name = best-practices). Candidate variants were ranked by filtering out common polymorphisms reported in public databases [Human Gene Mutation Database (HGMD), NHLBI GO Exome Sequencing Project (ESP), 1000 Genomes Project, Exome Aggregation Consortium (ExAc), database SNP (dbSNP)] or seen in the 10,681 exomes sequenced at Institut Imagine from families affected with genetic diseases. PolyPhen-2, SIFT and Mutation Taster were used to predict consequences of mutations on protein function. Mutations were next ranked on the basis of the predicted impact of each variant by combined annotation-dependent depletion (CADD), and compared with the mutation significance cut-off (MSC), a gene-level-specific cut-off for CADD scores (Itan et al, 2016).

### Sanger sequencing

ALPI variants identified by WES were confirmed by Sanger sequencing of PCR products amplified from genomic DNA. Primers used for DNA amplification were as follows: A97T mutation: forward primer, 5′-GGACCTTCAGTGGTTCCAGG-3′ and reverse primer, 5′-CCAAGGACCTGGTTCTGGTC-3′; A350V and A360V mutations: forward primer, 5′-CGCGGCTTCTACCTCT TTGT-3′ and reverse primer, 5′-GCCTACCGAAGATGGAGCTC-3′, and Q439X mutation: forward primer, 5′-GAGCGGTGAGTGA GGCTGAA-3′ and reverse primer, 5′-AGGTGGGGAGCAGGATAA CTC-3′. PCR was performed with AmpliTaq polymerase (Thermo Fisher Scientific, Villebon-sur-Yvette, France), using a GeneAmp PCR system (9700; Applied Biosystems, Thermo Fisher Scientific). PCR products were purified using QIAquick® Gel Extraction Kit (Qiagen) and sequenced using the same primers by Eurofins (the Genomic Platform of Université Paris Descartes, Paris, France).

### Sequence analysis and molecular modelling of ALPI 3D structure

The 3D model of human ALPI was built from the crystal structure of human placental alkaline phosphatase (entry code 1ew2; Le Du et al, 2001). Sequence homology between placental and intestinal alkaline phosphatases was assessed by alignment using the Basic Local Alignment Search Tool, BLAST (Altschul et al, 1997; Appendix Fig S1). The full-atom model of ALPI was next calculated with the program MODELLER (Sánchez & Sali, 2000). Conservation analysis was then performed based on our 3D model of ALPI using the fully automated evolutionary approach as implemented on ConSurf server (Ashkenazy et al, 2016). Finally, visualization, analysis and figure preparation of the 3D model were performed using PyMol software (DeLano, W.L. The PyMOL Molecular Graphics System 2002).

### Wild-type and mutant ALPI overexpression in HEK293T cells

For in vitro analysis of WT and mutant ALPI expression and activity, a vector (pCR4-TOPO) containing the full cDNA sequence of human ALPI transcript variant 1 (NM_001631.4; Dharmacon, Orsay, France) was used. p.Ala97Thr (c.289G>A, GCT to ACT), p.Ala350Val (c.1049C>T, GCG to GTG), p.Ala360Val (c.1079C>T, GCG to GTG), p.Glu439X (c.1315C>T, CAG to TAG) mutations were introduced using the GENEART® Site-directed Mutagenesis System

(Invitrogen, Thermo Fisher Scientific) according to manufacturer's instructions for annealing, extension times and cycle numbers. Primer sequences were as follows: 289_for 5′-CGCTTCCCATACCT GACTCTGTCCAAGACAT-3′;289_rev 5′-ATGTCTTGGACAGAGTCAG GTATGGGAAGCG-3′; 1049_for 5′-AGGCACTCACTGAGGTGGTCAT GTTCGACGA-3′, 1049_rev 5′-TCGTCGAACATGACCACCTCAGTGA GTGCCT-3′;1079_for 5′-ACGCCATTGAGAGGGTGGGCCAGCTCACC AG-3′, 1079_rev 5′-CTGGTGAGCTGGCCCACCCTCTCAATGGCGT-3′; 1315_for 5′-GGGAGCCCCGATTACTAGCAGCAGGCGGCGG-3′,1315_ rev 5′-CCGCCGCCTGCTGCTAGTAATCGGGGCTCCC-3′. Each construct was then fully sequenced and subcloned into the pLenti-CMV-GFP-2A-Puro plasmid (Applied Biological Materials, Richmond, CA). Lentiviral particles encoding the different mutants were generated by transfecting HEK293T cells with transfer plasmid, packaging expressing plasmid psPAX2 (Addgene, Cambridge, USA), VSV-G envelope expressing plasmid pMD2.G (Addgene) using Lipofectamine 2000 (Invitrogen, Thermo Fisher Scientific). Twelve hours after transfection, cells were washed, and fresh medium without antibiotics was added for 60 h. The recombinant virus-containing medium was filtered and used to transduce HEK293T cells in the presence of polybrene (4 μg/ml). Positively transduced cells were selected with puromycin (Gibco, Thermo Fisher Scientific) at 20 μg/ml concentration until all cells were GFP positive as assessed by FACS. Comparable transduction efficiency for WT and mutant constructs was ascertained based on fluorescence of the GFP reporter.

### Quantitative PCR

Total RNA was extracted from transduced-HEK293T cells using the RNeasy Plus extraction kit (Qiagen). Total RNA (5 μg) was reverse transcribed using M-MLV reverse transcriptase (Invitrogen). qRT–PCR was performed with an Assays-on-Demand probe (Applied Biosystems, Thermo Fisher Scientific) specific for ALPI-FAM (Hs00357579_g1) and RPLPO-FAM (Large Ribosomal Protein; Hs99999902_m1), which was used for normalization. qPCR was performed with a ABI PRISM 7900 (Applied Biosystems). Results are expressed according to the $\Delta\Delta C_t$ (cycle threshold) method.

### Cell lysis, immunoblotting and flow cytometry

Total proteins were solubilized in RIPA buffer (Sigma, Saint Quentin Fallavier, France) supplemented with 1× proteinase inhibitor cocktail mix (Roche, Sigma). Following separation by SDS–PAGE, immunoblotting was performed using Ab against ALPI (ab54776, 1 μg/ml, Abcam, Paris, France) and GAPDH (3683, 1:2,000, Cell Signaling). Surface cell staining of transduced cell lines was performed according to standard flow cytometry protocol using primary Ab against ALPI (ab54776, 1 μg/ml, Abcam), and secondary BV510 goat anti-mouse (Poly4053, 1:100, Sony Biotechnologies, Weybridge, UK) was used. Data were analysed on FACS Canto II (BD Biosciences, Rungis, France) using FlowJo software.

### Alkaline phosphatase activity

Alkaline phosphatase activity was measured in cell lysates of HEK293T cells expressing WT and mutant forms of ALPI by

**The paper explained**

**Problem**

Monogenic disorders associated with IBD-like intestinal inflammation are a group of severe and rare conditions. Identifying the molecular defect is necessary to optimize therapy and to improve long-term outcome. It also provides insights into key mechanisms indispensable for establishing and maintaining intestinal homeostasis in humans.

**Results**

Using whole-exome sequencing, we uncovered compound missense mutations in intestinal phosphatase alkaline, ALPI, in two unrelated patients. Severe colonic inflammation was associated with caeliac disease in one HLA-DQ2[+] child and with arthritis in the second patient. ALPI is a brush border metalloenzyme that catalyses phosphate hydrolysis of the lipid moiety of LPS and thereby drastically reduces LPS pro-inflammatory activity. Structural modelling and heterologous expression of ALPI mutants in HEK293T cells showed that all mutations were loss of function by impairing either stability or catalytic activity of ALPI. Decreased ALPI expression on small intestinal biopsies of both patients and lack of stool ALPI activity confirmed *in vitro* results.

**Impact**

This is the first time that loss-of-function mutations in ALPI are associated with monogenic forms of IBDs. Our results indicate that ALPI is one conserved regulatory mechanism, which plays a key role to limit inflammatory responses induced by microbiota-derived endotoxins. ALPI mutations should be screened for in individuals with inflammatory bowel disease refractory to treatment. Early diagnosis of ALPI deficiency should enable ALPI supplementation to prevent development of irreversible lesions and colectomy. ALPI supplementation might also be useful to alleviate inflammation in polygenic IBD patients, who usually display reduced ALPI expression.

assessing p-nitrophenyl phosphate (pNPP) dephosphorylation using Alkaline Phosphatase Assay Kit (ab83369, Abcam) according to the manufacturer's instructions.

**Alkaline phosphatase activity in stool**

Stool samples were weighed, resuspended in assay buffer (ab83369, Abcam) and homogenized in a FastPrep tube with six 2.5-mm glass beads installed (6913, MP Biomedicals, Illkirch, France) using the FastPrep®-24 Instrument (MP Biomedicals). Following centrifugation at 10,000 $g$ for 10 min, the supernatant containing AP was collected and assayed for AP concentration using Alkaline Phosphatase Assay Kit (ab83369, Abcam). Pre-treatment for 10 min with L-phenylalanine (L-Phe, 10 mM; P2126, Sigma), a specific inhibitor of ALPI or L-homoarginine (L-Arg, 10 mM; 101542, Sigma) a specific inhibitor of tissue-non-specific alkaline phosphatase (TNAP), was used to determine the major isoform among stool APs. The stool AP values are expressed as units of ALPI/g stool.

**LPS dephosphorylation activity**

To measure the LPS dephosphorylating activity of ALPI, HEK293T cells expressing WT and mutant forms of ALPI were first incubated

with LPS 100 ng/ml. Supernatants were collected and used to stimulate the human monocytic cell line THP1. Total RNA was then extracted from THP1 cells, and qRT–PCR was performed with an Assays-on-Demand probe (Applied Biosystems) specific for IL8-FAM (Hs00174103_m1) and RPLPO-FAM (Large Ribosomal Protein; Hs99999902_m1), for normalization.

**Immunohistochemistry, antibodies and confocal microscopy**

Formalin-fixed paraffin-embedded (FFPE) sections (5 μm) were cut and dewaxed by heating at 60°C and xylene for 10 min three times followed by different concentrations of ethanol to be rehydrated. Afterwards one set of slides was stained with haematoxylin and eosin for general views of histology. Antigen retrieval was performed with high pressure cooking in ethylenediaminetetraacetic acid (EDTA)-borate buffer [1 mM EDTA, 10 mM borax (sodium tetraborate, Sigma-Aldrich Co., St Louis, USA), 10 mM boric acid (Sigma-Aldrich Co.) with 0.001% Proclin 300 (Supelco, Bellefonte, USA)] at pH 8.5. The slides were then blocked for 1 h at room temperature with 4% BSA and 20% donkey serum in PBS, without $Ca^{2+}$ and $Mg^{2+}$. Incubation with mixed primary antibodies, rabbit polyclonal against ALPI Ab (PA5-22210, 1:200, Thermo Fisher, Burlington, Canada), mouse monoclonal against TNAP Ab (ab126820, 1:200, Abcam) and a goat polyclonal against E-cadherin Ab (AF748, 1:400, R&D System, Minneapolis, USA) was performed overnight at 4°C.

Slides were washed and incubated with donkey anti-rabbit IgG H+L-rhodamine conjugate (705-095-147, Jackson ImmunoResearch Waltham, USA) and donkey anti-goat IgG H+L-FITC conjugate (711-295-152, Jackson ImmunoResearch Waltham) for 2 h at room temperature in darkness. RedDot2 (1:200, Biotium, Hayward, USA) was used at a dilution of 1:200 for nuclear counter staining. These sections were mounted overnight with Vectashield fluorescence mounting medium (Vector Labs, Burlingame, USA).

Fluorescent confocal images of normal, IBDs and mutant duodenums or ileums for ALPI, TNAP and E-cadherin double-stained sections were obtained using a Leica confocal laser scanning microscope (model TCS-SPE) and LAS-AF software (Leica Microsystems Leica Microsystems, Wetzlar, Germany). Image processing was carried out with Adobe Photoshop CS3 software. To visualize these images, some pseudocolour such as far-red in blue was applied with the channel option of Photoshop CS3.

**Statistical analysis**

The statistical analysis was performed using GraphPad Prism version 6.0. Data are presented as mean ± standard deviation unless specified otherwise. Nonparametric, unpaired two-tailed Mann–Whitney test was used. Numbers of experiments and *P*-values are indicated in each figure legend.

**Data availability**

Whole-exome sequencing data have been deposited to the EGA database (https://www.ebi.ac.uk/ega/) with the assigned identifier EGAS00001002847 for patient P1 and EGAS00001002902 for patient P2.

**Expanded View** for this article is available online.

## Acknowledgements

This work was supported by Institutional grants from INSERM, by the European grant ERC-2013-AdG-339407-IMMUNOBIOTA, by the Investissement d'Avenir grant ANR-10-IAHU-01, by the Fondation Princesse Grace, by Fondation Maladies Rares. NCB benefits of an AP-HP Contrat d'Interface. AMM is supported by Leona M. and Harry B. Helmsley Charitable Trust and the Canadian Institute of Health Research (CIHR).

## Author contributions

Study concept and design: MP, NC-B; patient care coordination and clinical samples acquisition: FC-H, PF, NK, AMG, CR, FR, KF, PDA; acquisition of data: MP, FC-H, PF, RD-L, BB, JB, M-HLD, SR, JP, EC, NW, NG, MB, MZ; analysis and interpretation of data: MP, FC-H, M-HLD, NW, JB; drafting of the manuscript: MP, NW, AMM, NC-B; obtained funding: AMM, NC-B; study supervision: AMM, NC-B.

## Conflict of interest

The authors declare that they have no conflict of interest.

## For more information

Online Mendelian Inheritance in Man (OMIM), http://www.omim.org/; UCSC Genome Browser, http://genome.ucsc.edu/; PolyPhen-2, http://genetics.bwh.harvard.edu/pph2/index.shtml; SIFT, http://sift.bii.a-star.edu.sg/; Mutation Taster, http://www.mutationtaster.org/; Human Gene Mutation Database, http://www.hgmd.cf.ac.uk/ac/index.php, NHLBI GO Exome Sequencing Project (ESP), http://evs.gs.washington.edu/EVS/; 1000 Genomes Project, http://www.internationalgenome.org/data; dbSNP, https://www.ncbi.nlm.nih.gov/projects/SNP/; Exome Aggregation Consortium (ExAC) Browser, http://exac.broadinstitute.org/; The Mutation Significance Cut-off (MSC) Server, http://pec630.rockefeller.edu:8080/MSC/; Clustal Omega, https://www.ebi.ac.uk/Tools/msa/clustalo/.

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
