## [Review Process File · EMBO Molecular Medicine]

Human ALPI deficiency causes inflammatory bowel disease and highlights a key mechanism of gut homeostasis

Marianna Parlato, Fabienne Charbit-Henrion, Jie Pan, Claudio Romano, Rémi Duclaux-Loras, Marie-Helene Le Du, Neil Warner, Paola Francalanci, Julie Bruneau, Marc Bras, Mohammed Zarhrate, Bernadette Bègue, Nicolas Guegan, Sabine Rakotobe, Nathalie Kapel, Paola De Angelis, Anne M Griffiths, Karoline Fiedler, Eileen Crowley, Frank Ruemmele, Aleixo M. Muise, Nadine Cerf-Bensussan

Review timeline:

Submission date:	13 September 2017
Editorial Decision:	12 October 2017
Revision received:	4 January 2018
Editorial Decision:	31 January 2018
Revision received:	19 February 2018
Accepted:	19 February 2018

Editor: Céline Carret

Transaction Report:

1st Editorial Decision

12 October 2017

Thank you for the submission of your manuscript to EMBO Molecular Medicine. We have now heard back from the two referees whom we asked to evaluate your manuscript.

As you will see from their comments below, the referees find the study of interest. They highlight a few issues however that must be addressed. During our cross-commenting exercise, it appears that improving the clinical significance of the data by examining ALPI isoforms levels and activities in the stool of probands seems to be one of the most important concern. Another approach for a similar end would be to determine whether or not LPS activity is decreased in the stool of the probands, given that that is the proposed mechanism by which ALPI protects against IBD. Equally suggested and relevant to strengthen the paper would be to explain the role of "unresponsive" macrophages in disease progression. Finally, both referees request additional and more thorough discussion of the findings.

We would welcome the submission of a revised version within three months for further consideration and would like to encourage you to address all the criticisms raised as suggested to improve conclusiveness and clarity. Please note that EMBO Molecular Medicine strongly supports a single round of revision and that, as acceptance or rejection of the manuscript will depend on another round of review, your responses should be as complete as possible.

I look forward to receiving your revised manuscript.

***** Reviewer's comments *****

Referee #1 (Comments on Novelty/Model System for Author):

This is potentially an important discovery and suggests that IAP could play an important role in

human IBD and also provides support for using IAP as a therapy.

Referee #1 (Remarks for Author):

This is a very interesting observation and discovery regarding IAP. The data lend strong support to the idea that this enzyme plays a role in the pathogenesis of some human IBD and also that it could be a good therapy for IBD. There are some issues that should be addressed by the authors.

1) In mice, loss of the major form of IAP (AKP3 KO mice) leads to an increase in the expression of other forms of IAP (AKP 5 and 6). it would be of interest therefore to determine whether the decrease IAP expression in these patients is associated with any increase in another form of AP from their guts.

2) Related to point #1, it would be of interest to measure the enzyme activity of AP and IAP specifically in the stool of these patients and compare to other patients with and without IBD. Do we know whether overall IAP activity in the luminal contents is decreased in these patients? Could stool IAP activity be a useful biomarker for susceptibility to IBD?

3) The two patients exhibit very different clinical courses and phenotypes in regard to their IBD. Wouldn't one expect a more similar form of IBD in these patients if their important underlying abnormality is related to the same gene product?

4) On page 15 the authors indicate that these mutations are not just a chance finding. Can they be more quantitative in that assessment and provide the percent chance that such mutations are not related to the IBD in the two patients?

On page 17, requested should be required.

5) There is no discussion of any limitations to this study.

Referee #2 (Remarks for Author):

Parlato et al provide convincing evidence that the two probands suffering from IBD are compound heterozygotes for variant intestinal alkaline phosphatase alleles that result in loss of function of the protein and a consequent inability to detoxify LPS molecules.

They outline a body of evidence that suggests that alkaline phosphatase has an important role in gut homeostasis and innate immunity by dephosphorylating toxic microbial ligands, including LPS, and protecting the host from uncontrolled inflammatory responses (ie, IBD, but also necrotizing enterocolitis). Most of this work comes from experimental models in rodents, but there is a paper by Lukas et al (number 31 in reference list) from 2010 suggesting that enteral alkaline phosphatase is beneficial in patients with ulcerative colitis when given by a constant infusion through a naso-duodenal tube for 7 days. My literature review has not demonstrated further progress with this therapeutic approach, but the possible mechanism of action of exclusive enteral nutrition (polymeric formula) was suggested to be induction of small intestinal ALPI production on intestinal cell lines (Caco-2). [Budd et al, *Innate Immun.* 2017 Apr;23(3):240-248. doi: 10.1177/1753425916689333. Epub 2017 Jan 19.] There is also evidence that inflammation (eg. IBD or celiac disease), results in decreased small intestinal expression of ALPI.

I would find the links between IBD, innate immunity, gut homeostasis and ALPI more compelling if the authors could explain how human (murine and rat) ALPI, which is expressed at very high levels in the small intestine and at very low levels in the colon (Tuin et al, reference 18, and the database, Bio-GPS), protects against colonic inflammation. This is particularly relevant since the colonic bacterial load is orders of magnitude higher than in the small intestine.

The other major unaddressed observation is that intestinal macrophages express low levels of bacterial response elements, including the TLRs, and are generally regarded as being unresponsive to bacterial products in the colonic microenvironment in any case. How does detoxifying bacterial products (presumably in the small bowel) protect against the development of colitis in this context. Similarly, with colonic epithelial TLR expression.

Proband 1 has the phenotype of very early onset IBD, but proband 2 does not. I imagine that the exome sequencing was done first on P1 and that the "causative" alleles identified. It is difficult to assess the frequency of functional variants in ALPI in the general population, but how many IBD

patients were screened before variants in ALPI were found in a second patient?

It is, of course, possible that the variants in ALPI are the cause of IBD in these two patients, but that there are other mechanisms by which IBD develops. In my opinion, the Discussion should reflect this. But again, there is the difficulty in explaining how a small intestinal protein protects against colonic disease.

1st Revision - authors' response

4 January 2018

Referee #1 (Comments on Novelty/Model System for Author):

This is potentially an important discovery and suggests that IAP could play an important role in human IBD and also provides support for using IAP as a therapy.

Referee #1 (Remarks for Author):

This is a very interesting observation and discovery regarding IAP. The data lend strong support to the idea that this enzyme plays a role in the pathogenesis of some human IBD and also that it could be a good therapy for IBD. There are some issues that should be addressed by the authors.

We thank the reviewer of her/his very positive comments.

1) In mice, loss of the major form of IAP (AKP3 KO mice) leads to an increase in the expression of other forms of IAP (AKP 5 and 6). it would be of interest therefore to determine whether the decrease IAP expression in these patients is associated with any increase in another form of AP from their guts.

1) Answer: We thank the reviewer for this pertinent suggestion and have now performed the requested experiment.

In humans, four isoforms of ALPs have been identified: placental alkaline phosphatase (PLAP), intestinal alkaline phosphatase (ALPI), tissue-nonspecific alkaline phosphatase (TNAP), germ cell ALP (GALP). Given the tissue specificity of PLAP and GALP, we focused our complementary analysis on TNAP. As now discussed in the new paragraph in the results section (lines 238-241) and shown in Fig. EV4, "TNAP staining largely predominated in lamina propria in controls and in ALPI-deficient patients. Yet, P1 and P2 also displayed some epithelial expression, suggesting that TNAP might perhaps compensate for ALPI deficiency in the intestinal lumen". As suggested by the reviewer in his/her second related comment, this point was addressed by assessing AP activity in stools. We did not find evidence of ALPI or TNAP activity in the stools of P1 and therefore concluded that there was no compensatory role by TNAP in the absence of ALPI. This point is more precisely discussed in the answer to the second and related comment of reviewer 1.

2) Related to point #1, it would be of interest to measure the enzyme activity of AP and IAP specifically in the stool of these patients and compare to other patients with and without IBD. Do we know whether overall IAP activity in the luminal contents is decreased in these patients

2) Answer: As indicated lines 242-255, "AP activity was assessed in stools in the presence of l-phenylalanine (l-Phe), a specific inhibitor of ALPI or of l-homoarginine (l-Arg), a specific inhibitor of TNAP. As shown in Fig 4C, stools of non-inflamed controls displayed substantial AP activity, which was largely inhibited by l-Phe but not by l-Arg. This result was consistent with previous reports indicating that most AP activity in stools is due to ALPI with some residual AP activity of microbial origin (Malo MS, 2015) (Fig 4C). ALPI activity was significantly reduced in stools of patients with intestinal inflammation (fecal calprotectin >250 mg/g), a result in keeping with previous studies showing decreased ALPI expression in the inflamed intestine (Molnár K et al, 2012a; Molnár K et al, 2012b). However, TNAP remained, as in controls, undetectable. ALPI analysis could not be performed in P2, who refused further stool sampling, but both ALPI and TNAP were undetectable in P1 stools sampled at time of remission two years after colectomy, altogether confirming the loss-of-function mutations in ALPI and the lack of intraluminal substitution by TNAP in this patient."

Could stool IAP activity be a useful biomarker for susceptibility to IBD?

As now shown in figure 4D, ALPI activity was significantly decreased in patients with intestinal inflammation (defined by calprotectin > 250 mg/mL) compared to non-inflamed controls (defined by calprotectin < 250 mg/mL). Yet, ALPI activity was very variable in non-inflamed controls with a large overlap of values with those observed in stools from inflamed patients. Although the number of tested patients is small, we suggest that ALPI activity is unlikely to be a robust biomarker of intestinal inflammation.

3) The two patients exhibit very different clinical courses and phenotypes in regard to their IBD. Wouldn't one expect a more similar form of IBD in these patients if their important underlying abnormality is related to the same gene product?

3) Answer: The reviewer raises an important issue. Yet, with the development of next generation sequencing, there are more and more examples of mutations in the same gene resulting in a spectrum of phenotypes and variable penetrance. This may be more particularly true for genes regulating immune intestinal responses to environmental factors, as it is likely the case for ALPI. Thus patients with LRBA loss of function, NCLR4 and STAT3 gain of function or CTLA4 haploinsufficiency display variable intestinal phenotype and very variable age at onset. Whether and how ALPI deficiency is instrumental in predisposing P1 and P2 to severe intestinal inflammation is now more thoroughly discussed lines 270-302 (in red). Whether or not ALPI deficiency might promote autoimmune-like manifestations is also discussed more carefully lines 302-314.

4) On page 15 the authors indicate that these mutations are not just a chance finding. Can they be more quantitative in that assessment and provide the percent chance that such mutations are not related to the IBD in the two patients?

As shown in table 1, the minor allele frequency (MAF) in the Exome Aggregation Consortium (ExAC) population dataset comprising exome sequencing data from 60,706 unrelated individuals was very low for the four reported loss of functions ALPI mutations as comprised between 0 and 0.00063 and no homozygous individuals for any of these mutations nor for any predicted loss-of-functions mutations (splice donor or acceptor, stop gained and frameshift) was present in ExAC. In contrast, the compound loss-of-function mutations reported here were found in 2 patients out of 1144 cases of paediatric IBD, a frequency much higher than the one predicted for the same compound mutations in the EXAC data (between 0 and 0.0000008). Yet, it would be interesting to identify other IBD patients with loss of function mutations in ALPI to strengthen the causality link.

On page 17, requested should be required.

We have now corrected it.

5) There is no discussion of any limitations to this study.

5) Answer: A first limitation of this study is the identification of only two patients with loss of function mutations in ALPI, making difficult to accurately define clinical consequences. This is now acknowledged lines 274 and 275. Yet, as emphasized in Casanova JL et al, 2014 “single-patient studies can be conclusive, provided there is rigorous selection of variations in silico followed by in-depth experimental validation in vitro via the dual characterization of the mutant alleles and a cellular or animal phenotype, which establishes a causal bridge between a candidate genotype and a clinical phenotype”. A second limitation concerns ALPI oral supplementation which seems to be an attractive therapeutic option. Yet, this treatment was not applicable to the patients since, at the time of molecular diagnosis, they were in remission after colectomy as now indicated lines 325-327.

Referee #2 (Remarks for Author):

Parlato et al provide convincing evidence that the two probands suffering from IBD are compound heterozygotes for variant intestinal alkaline phosphatase alleles that result in loss of function of the protein and a consequent inability to detoxify LPS molecules.

We thank the reviewer for her/his very positive comments.

They outline a body of evidence that suggests that alkaline phosphatase has an important role in gut homeostasis and innate immunity by dephosphorylating toxic microbial ligands, including LPS, and protecting the host from uncontrolled inflammatory responses (ie, IBD, but also necrotizing enterocolitis). Most of this work comes from experimental models in rodents, but there is a paper by Lukas et al (number 31 in reference list) from 2010 suggesting that enteral alkaline phosphatase is beneficial in patients with ulcerative colitis when given by a constant infusion through a naso-duodenal tube for 7 days. My literature review has not demonstrated further progress with this therapeutic approach, but the possible mechanism of action of exclusive enteral nutrition (polymeric formula) was suggested to be induction of small intestinal ALPI production on intestinal cell lines (Caco-2). [Budd et al, Innate Immun. 2017 Apr;23(3):240-248. doi: 10.1177/1753425916689333. Epub2017 Jan19.]

There is also evidence that inflammation (eg, IBD or celiac disease), results in decreased small intestinal expression of ALPI.

I would find the links between IBD, innate immunity, gut homeostasis and ALPI more compelling if the authors could explain how human (murine and rat) ALPI, which is expressed at very high levels in the small intestine and at very low levels in the colon (Tuin et al, reference 18, and the database, Bio-GPS), protects against colonic inflammation. This is particularly relevant since the colonic bacterial load is orders of magnitude higher than in the small intestine.

Answer: We thank the reviewer for this remark which has allowed us to stress that ALPI, indeed mainly expressed in the small intestinal brush border, is released into the lumen within vesicles that transport the enzyme under its active form toward distal intestinal sites (Tuin A *et al*, 2009; Shifrin DA Jr *et al*, 2012; Bates JM *et al*, 2007; Malo MS *et al*, 2015; Eliakim R *et al*, 1991) (Yang WH *et al*, 2017). This indication is now given lines 228-231 in the results section and lines 262-265 in the discussion.

The other major unaddressed observation is that intestinal macrophages express low levels of bacterial response elements, including the TLRs, and are generally regarded as being unresponsive to bacterial products in the colonic microenvironment in any case. How does detoxifying bacterial products (presumably in the small bowel) protect against the development of colitis in this context. Similarly, with colonic epithelial TLR expression.

Answer: We thank the reviewer for this pertinent comment. We have now modified the discussion as follows lines 283-299: “In mammals, there is ample evidence that continuous intestinal exposure to LPS induces the degradation of signalling molecules downstream TLR4, so that, at steady state, epithelial cells and lamina propria mononuclear phagocytes are largely unresponsive to LPS (Smith PD *et al*, 2001; Lotz M *et al*, 2006). Endogenous tolerance to LPS is however broken upon induction of inflammation induced either by gut injury or infection, notably due to the recruitment of inflammatory monocytes into the intestinal mucosa (Kamada N *et al*, 2008). ALPI-mediated detoxification may then become indispensable to curb inflammation. This scenario is coherent with data in ALPI-deficient mice, which did not spontaneously develop intestinal inflammation, but displayed markedly increased susceptibility to dextran sodium-sulfate-induced colitis (Ramasamy S *et al*, 2001) as well as increased gut permeability (Rentea RM *et al*, 2012) and enhanced LPS translocation when subjected to direct or indirect gut injury (Kaliannan K *et al*, 2013). This scenario is also supported by data in mice submitted to recurrent non-lethal infection by *S. typhimurium*, which eradicate the bacteria but finally develop chronic colitis due to increased degradation of ALPI and lesser availability of active ALPI in the intestinal lumen (Yang WH *et al*, 2017)”.

Proband 1 has the phenotype of very early onset IBD, but proband 2 does not. I imagine that the exome sequencing was done first on P1 and that the "causative" alleles identified. It is difficult to assess the frequency of functional variants in ALPI in the general population, but how many IBD patients were screened before variants in ALPI were found in a second patient?

Answer: The first patient was identified in by WES performed in 55 patients with VEOIBD. The second patient was identified by screening 1089 WES performed in an independent cohort of pediatric IBD patients (Canada).

It is, of course, possible that the variants in ALPI are the cause of IBD in these two patients, but that there are other mechanisms by which IBD develops. In my opinion, the Discussion should reflect this. But again, there is the difficulty in explaining how a small intestinal protein protects against colonic disease.

Answer: As indicated above, we have now better explained how ALPI produced in the small intestine can exert its enzymatic activity at distant sites. We have also extensively modified the discussion to examine whether and how ALPI deficiency may contribute to the clinical phenotypes in P1 and P2 (lines 262-299).

2nd Editorial Decision

31 January 2018

Thank you for the submission of your revised manuscript to EMBO Molecular Medicine. We have now received the enclosed reports from the referees that were asked to re-assess it. As you will see the reviewers are now globally supportive and I am pleased to inform you that we will be able to accept your manuscript pending a few final amendments:

1) Please address the referees' comments in writing. At this stage, we'd like you to discuss referee's 2 point on stoichiometry and whether there is enough alkaline phosphatase activity to neutralise the vast quantities of LPS in the colon.

***** Reviewer's comments *****

Referee #1 (Comments on Novelty/Model System for Author):

This is the first report of a human mutation in ALP1 associated with a disease. Very interesting.

Referee #1 (Remarks for Author):

There are a few grammatical errors. The manuscript should be carefully edited to correct these.

Referee #2 (Remarks for Author):

My questions have been addressed well.

Do the authors have any sense of the stoichiometry? How much alkaline phosphatase is needed to inactivate how much bacterial LPS in the colon?

2nd Revision - authors' response

19 February 2018

Referee #1 (Comments on Novelty/Model System for Author):

This is the first report of a human mutation in ALP1 associated with a disease. Very interesting.

Referee #1 (Remarks for Author):

There are a few grammatical errors. The manuscript should be carefully edited to correct these.

We thank the reviewer for his/her positive appreciation of our work. We have now edited the manuscript to correct grammatical errors.

Referee #2 (Remarks for Author):

My questions have been addressed well.

We thank the reviewer for his/her positive appreciation of the work performed to answer his/her comments

Do the authors have any sense of the stoichiometry? How much alkaline phosphatase is needed to inactivate how much bacterial LPS in the colon?

We do not have experimental data regarding the exact stoichiometry ALPI-LPS in the colon. However, as ALPI secreted in the stools is likely the result of its production and degradation in the gut, we can speculate that about 300 U/g stool, the amount of ALPI we found in stools collected from not inflamed individuals is enough to inactivate LPS in the colon and maintain homeostasis.

Corresponding Author Name: Nadine Cerf-Bensussan and Aleixo Muise

Journal Submitted to: EMM

Manuscript Number: EMM-2017-08483